# Unconventional polarization fatigue in van der Waals layered ferroelectric ionic conductor $CuInP_2S_6$

Ziwen Zhou[1,4], Shun Wang[1,4], Zhou Zhou[1], Yiqi Hu[1], Qiankun Li[1], Jinshuo Xue[1], Zhijian Feng[1], Qingyu Yan[1], Zhongshen Luo[1], Yuyan Weng[1]✉, Rujun Tang[1], Xiaodong Su[1], Fengang Zheng[1], Kazuki Okamoto[2], Hiroshi Funakubo[2], Lixing Kang ◉[3], Liang Fang ◉[1]✉ & Lu You ◉[1]✉

Recent progress in two-dimensional ferroelectrics greatly expands the versatility and tunability in van der Waals heterostructure based electronics. However, the switching endurance issue that widely plagues conventional ferroelectrics in practical applications is hitherto unexplored for van der Waals layered ferroelectrics. Herein, we report the observation of unusual polarization fatigue behaviors in van der Waals layered $CuInP_2S_6$, which also possesses finite ionic conductivity at room temperature. The strong intertwinement of the short-range polarization switching and long-range ionic movement in conjunction with the van der Waals layered structure gives rise to unique morphological and polarization evolutions under repetitive electric cycles. With the help of concerted chemical, structural, lattice vibrational and dielectric analyses, we unravel the critical role of the synergy of ionic migration and surface oxidation on the anomalous polarization enhancement and the eventual polarization degradation. This work provides a general insight into the polarization fatigue characteristics in ionically-active van der Waals ferroelectrics and delivers potential solutions for the realization of fatigue-free capacitors.

Ferroelectric fatigue, manifested by the loss of switchable polarization under repetitive bipolar electric cycles, is a long-standing issue that critically impedes the practical applications of ferroelectric materials in reliable logic and memory devices[1,2]. Ever since the first report of ferroelectric fatigue phenomenon in $BaTiO_3$ by Merz and Anderson[3], numerous efforts were devoted to understanding its microscopic mechanisms[4–6], and thereupon exploring potential remedies[7–9]. Most existing models were established on oxide ferroelectrics, in which near-electrode charge injections and oxygen vacancy redistributions are extensively involved[10–13]. Recent research fevor in two-dimensional

(2D) materials rejuvenated considerable interest in van der Waals (vdW) layered ferroelectrics, primarily composed of chalcogenides and halides[14–18]. The reduced lattice dimensionality and unique stacking degree of freedom bring about unconventional properties and physics in 2D vdW ferroelectrics, such as negative longitudinal piezoelectricity and sliding/moiré ferroelectricity[19,20]. Additionally, lower electronegativities of the chalcogens and halogens, compared to oxygen, usually result in weaker chemical bonds and potentially higher ionic conductivity in vdW layered ferroelectrics[21,22]. This results in anomalous switching characteristics, including multi-well polarization

[1]School of Physical Science and Technology, Jiangsu Key Laboratory of Thin Films, Soochow University, Suzhou 215006, China. [2]School of Materials and Chemical Technology, Tokyo Institute of Technology, Yokohama 226-8502, Japan. [3]Division of Advanced Materials, Suzhou Institute of Nano-Tech and Nano-Bionics, Chinese Academy of Sciences, Suzhou 215123, China. [4]These authors contributed equally: Ziwen Zhou, Shun Wang. ✉e-mail: wengyuyan@suda.edu.cn; lfang@suda.edu.cn; lyou@suda.edu.cn

states[23–25], ionic-defect-coupled polarization switching[26,27], and potentially quantum polarization switching[28,29]. The successful integration of 2D vdW ferroelectrics into innovative device paradigm pivots on the reliable and enduring switching of ferroelectric polarization. However, polarization fatigue characteristics in vdW layered ferroelectrics are hitherto unreported, and the role of finite ionic conduction in the fatigue process remains unknown. In a bid to throw light into these issues, we systematically investigated the polarization degradation behaviors of a prototypical vdW layered ferroelectric ion conductor, $CuInP_2S_6$ (CIPS), under repetitive electrical stress. Through field strength-, cycling frequency-, and temperature-dependent tests, we discovered unconventional fatigue characteristics of CIPS capacitors, such as protrusion formation, strong morphological rippling, and unconventional enhancement of the polarization. These unique features are closely linked to the layered structure and sizable $Cu^+$ ion activities of CIPS, as revealed by our correlative and comparative chemical, structural and local piezoelectric response measurements. This work delivers in-depth understanding of the microscopic mechanism of ferroelectric degradation process in vdW layered ferroelectrics, enabling future quests for fatigue-free vdW ferroelectrics.

## Results

### General fatigue behaviors of CIPS capacitors

CIPS single crystals with thicknesses ranging from approximately 10–16 μm were used to fabricate parallel plate ferroelectric capacitors with coercive fields ($E_C$) of ≈25–30 kV cm⁻¹ (See Table S1 for sample details). We first studied the polarization fatigue behaviors of CIPS under different field strengths (the pulse width and the switching frequency are fixed at 1 ms and 100 Hz, respectively), as summarized in Fig. 1. At relatively small fields (≈$1.7E_C$), the hysteresis loop shows a progressive increase of the switched polarization with cumulative cycles (Fig. 1b). Under higher stressing fields (≈$3.4E_C$ and $4.1E_C$), the switched polarization value also increases initially, but drops after certain cycles, accompanied by the enhancement of the leakage current as indicated by the vertical blown-up of the loops. Meanwhile, all samples show some increments in the coercive field. To accurately determine the switchable remanent polarization ($\Delta P_r$), positive-up-negative-down (PUND) method was employed to subtract the contributions from non-remanent polarization and leakage charges. As shown in Fig. 1e, CIPS capacitor stressed at higher field tends to fatigue faster after the polarization upturn, while those stressed at low field show anomalous increments of the remanent polarization up to $4 \times 10^6$

cycles. Such polarization enhancement is further verified by standard and remanent hysteresis loops as well as PUND measurements (Fig. S1).

During the electrical cycling process, the morphologies of the CIPS capacitors also experienced drastic changes. A typical morphological evolution under electric cycles is presented sequentially in Fig. 1a (see Fig. S2 for the dark-field and transmission images and Supplementary Movie 1). The general features at different stages can be summed up as follows. First, dark region started to appear around the electrode edge (red arrow) at about $10^5$ cycles. In the meantime, bubble-like protrusions gradually grew out under the electrode (blue arrow). Lastly, surface ripples begin to form in the surrounding area, resulting in Newton's rings due to the delamination of the vdW layers (yellow arrow). With incremental cycles, the aforementioned three features grow larger and denser. Besides, there is a general trend that the surface ripples produced by low-field stressing are smaller and denser, while high field promotes large-scale layer delamination (Fig. S3).

The frequency-dependent fatigue behavior was also studied under a constant field of about $2.8E_C$ as shown in Fig. S4. Similar to the field-dependent series, all the samples show a polarization upturn from $10^5$ to $10^6$ cycles. However, there is no clear dependence of the polarization degradation on switching frequency, except that the capacitors fatigued by lower repetition rate tends to show larger polarization enhancement with almost doubled $\Delta P_r$ (Fig. S1c, d). Overall, based on the field- and frequency-dependent studies, the CIPS capacitors under repetitive electrical switching cycles commonly exhibit an increase of the leakage, an enhancement of the polarization at the early stage and similar morphological evolution. However, the eventual polarization degradation occurs more in an erratic and stochastic manner, possibly because the enhanced leakage inhibits effective polarization switching. Our fatigue test on CIPS flake (≈200 nm) shows similar behaviors as the bulk single crystal (Fig. S5). Hence, in the following sections, our investigations are mainly focused on the bulk crystal.

### Correlated microscopic imaging of fatigued CIPS capacitors

To understand the microscopic origin of the unconventional fatigue behaviors, compositional analysis by energy dispersive X-ray spectroscopy (EDS) was performed to reveal possible changes of surface chemistry associated with the morphology modifications. By a direct comparison of elemental mapping between pristine and fatigued capacitors, large amount of oxygen is found to accumulate at the

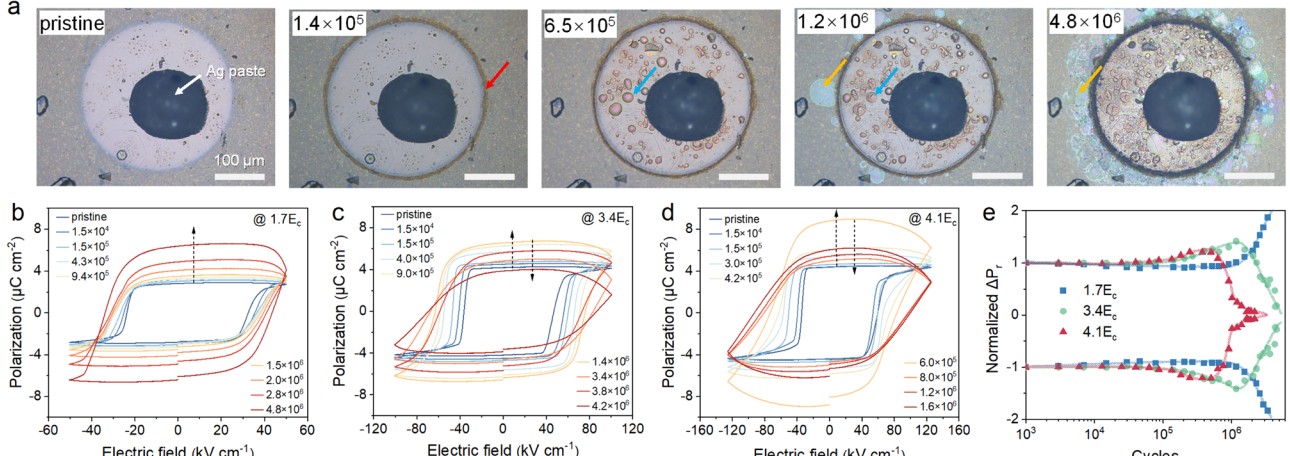

**Fig. 1 | Ferroelectric fatigue behavior of $CuInP_2S_6$ under different electric fields.** **a** Sequential optical images of the same CIPS capacitor after cumulative polarization switching cycles under an electric field of $3E_C$. The scale bar is 100 μm. **b**–**d** Evolutions of the polarization hysteresis loops of CIPS capacitors after cumulative cycles with an electric field of $2E_C$, $3E_C$ and $4.5E_C$, respectively. The switching frequency and pulse width is fixed at 100 Hz and 1 ms, respectively. **e** Switchable remanent polarization as a function of the switching cycles under different field strengths measured by PUND method.

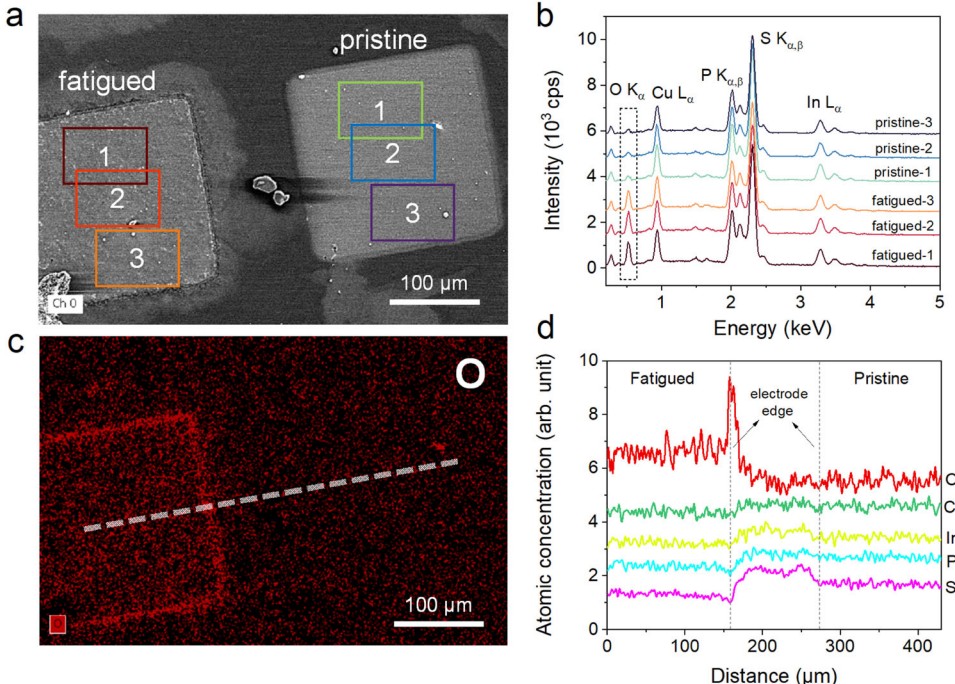

**Fig. 2 | Surface chemical changes in fatigued CuInP$_2$S$_6$. a** The SEM image of a pristine and a fatigued CIPS capacitor. **b** EDS spectra averaged over different areas marked by the corresponding color boxes in (**a**). The dash box highlights the stark difference at O K$_\alpha$ line. **c** The corresponding oxygen distribution map of (**a**). **d** EDS line profiles of different elements along the dash line shown in (**c**).

electrode edge of the fatigued sample (Fig. 2a, c). EDS spectra and corresponding quantitative analyses of the atomic compositions further indicate that the overall oxygen content underneath the electrode is also greatly increased in fatigued sample compared to the pristine one (Fig. 2b and Table S2). However, the relative percentage of the constituent elements of CIPS remain almost the same. By scrutinizing the line profiles of each element (Fig. S6), the increase of O concentration at the electrode edge is accompanied by the corresponding dips in the relative concentration of In, P and S as expected, because the total atomic concentration sums up to 100%. However, this case was not observed for Cu, suggesting a relative enrichment of Cu at the electrode edge (Fig. 2d).

Multiple microscopic imaging techniques were then employed to reveal the correlation between structure and property in fatigued capacitor with unconventional morphological changes, as summarized in Fig. 3. Two representative regions were investigated, namely, the electrode edge and the rippled area that spreads far way, as denoted by the blue and red boxes in Fig. 3a. Correlated with the optical images (Fig. 3d, j), the AFM topographic image (Fig. 3e) of the rippled area is featured by hillocks with tens to hundreds of nanometers in height, whereas the surface ripples near the electrode edge are so dense that they strongly merge with each other (Fig. 3k). The dark area at the electrode edge shown in optical image exhibits much higher outgrowth than the rippled area. Our EDS mapping of the same locations indicated obvious oxygen accumulation in the rippled area (Fig. 3f), which is even more intense around the electrode edge (Fig. 3l). Besides, we also observed slight Cu enrichment in the rippled area compared to the other three elements (Fig. S7), in agreement with the result of Fig. 2d. Furthermore, we also imaged the surface potential (Fig. 3g, m) and piezoelectric response (Fig. 3h, n) of these two areas using scanning Kelvin probe microscopy (SKPM) and piezoresponse force microscopy (PFM). The results indicated reduced surface potential and piezoelectricity in the rippled surface and electrode edge compared to those of the pristine surface. Raman spectra of selected spots (denoted by color circles in optical images) were recorded,

which identified characteristic peaks from cyclooctasulphur S$_8$ rings at the electrode edge (Fig. 3b)[30,31]. Its signature peak at 472 cm$^{-1}$ peak due to the symmetric S-S bond stretching is also visible in the Raman spectrum recorded at the rippled surface, suggesting its direct connection with the surface oxidation. Last but not least, conductive AFM (CAFM) was used to probe the conductivity changes caused by the potential oxidation in the rippled area and electrode edge. As the measurement on top of the Au electrode would result in huge currents that might damage the tip, we chose another dark region (marked by purple box in Fig. 3a) near the electrode edge for imaging. Interestingly, the regions with oxygen accumulation also possess higher electrical conductivity as evidenced by the CAFM images, and the conductivity seems to scale with the oxygen amount (Fig. 3i, o). The local current – voltage curves confirmed the strongly enhanced conductivity at the surface ripple than at pristine surface (Fig. 3c). All the above results are highly correlated, and point to the important implication that repetitive electric cycles cause substantial surface oxidation at the electrode edge and its surrounding areas. The oxidation results in significant changes in the morphological, structural and electronic properties. No detectable oxide peak was observed in the Raman spectra, suggesting the oxidized product is probably non-crystalline. Nevertheless, the oxidized surface layer possesses enhanced electrical conductivity, which may potentially increase the electrode area and the measured polarization.

In terms of the bubble-like protrusion on the electrode, additional experiments were carried out to elucidate its nature. First of all, we found the surface protrusions were considerably less under the silver paste or on the back side of the capacitor (Fig. S8). Interestingly, the protrusions behaved like a balloon showing repetitive inflations and evacuations without breaking up (Fig. S9 and Supplementary Movie 1). These observations made us to speculate that the protrusions are filled by gases instead of liquid or solid substances due to electrically induced phase decomposition1[2,32]. To verify our guess, we used microsized tungsten probe to scratch and open up the bubble (Fig. S10a–c and Supplementary Movie 2). The exposed surface is featured

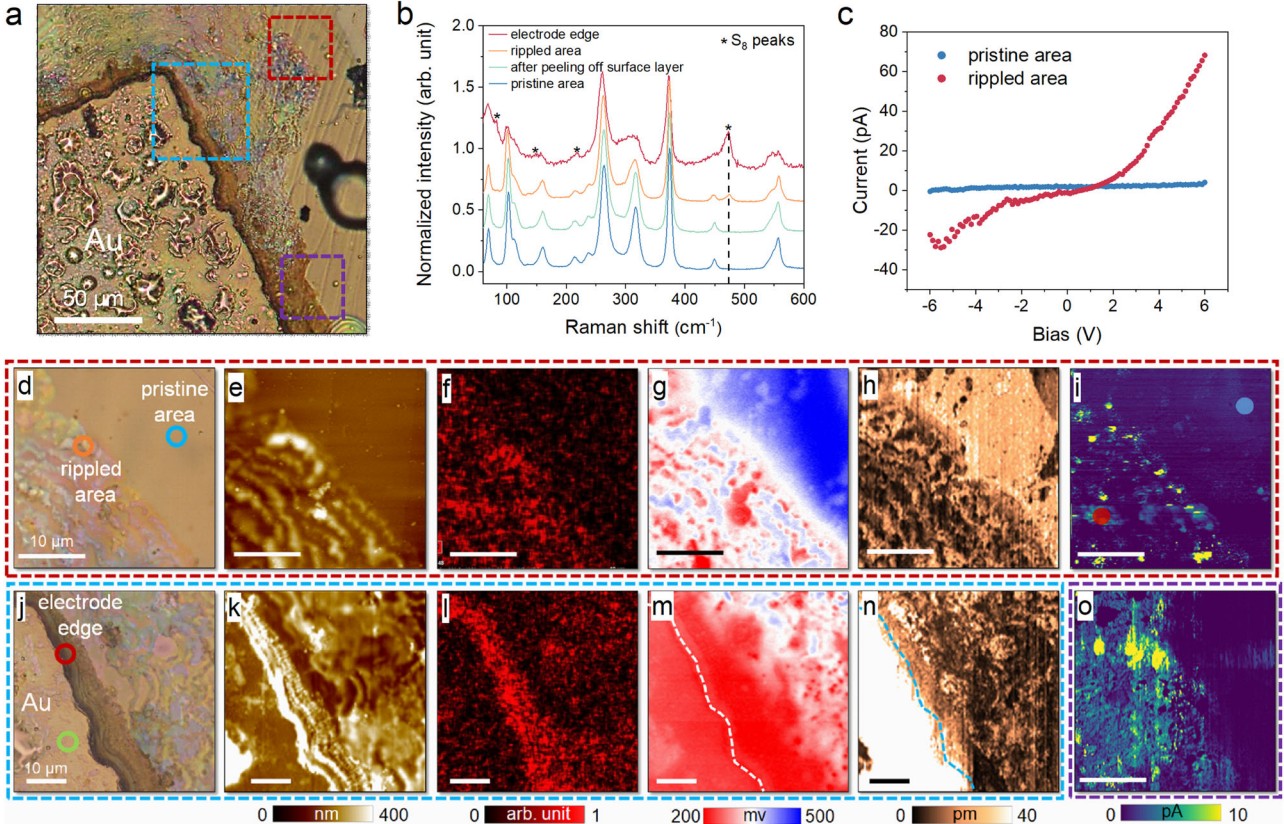

**Fig. 3 | Correlated microscopic imaging of the fatigued capacitor. a** Optical image of a fatigued CIPS capacitor. **b** Raman spectra collected at representative spots marked in (**d**) and (**j**). **c** Local current-voltage curves measured at pristine and rippled area marked in (**i**). Correlated microscopic images of the areas denoted by (**d–i**) red, (**j–n**) blue and (**o**) purple dash box shown in (**a**) using multiple imaging techniques: (**d**, **j**) optical images, (**e**, **k**) AFM topographic images, (**f**, **l**) O element maps, (**g**, **m**) SKPM images, (**h**, **n**) PFM images, and (**i**, **o**) CAFM images.

by ring-shape dark areas at the bubble edge (Fig. S10d). Subsequent Raman and EDS spectra confirm that the dark area inside the bubble also possesses oxygen enrichment and the characteristic peak of $S_8$ rings, suggesting similar surface oxidation as the electrode edge and rippled area (Fig. S10e–k). These results indicate the gas inside the bubbles are possibly air, and the progressive inflation of the bubbles results in ring-like oxidation dark lines at the electrode-sample-air interface.

**Depth-resolved compositional chemistry of the fatigued CIPS**
To uncover the compositional chemistry along the sample depth, depth-profiling X-ray photoelectron spectroscopy (XPS) was conducted on both pristine and fatigued capacitors. The sample details are shown in Fig. S11. The probed areas were located at the center of the capacitors with an X-ray beam size of 50 μm in diameter. The depth profiling was performed by sequential $Ar^+$ ion etching, in which the nominal thickness is estimated based on the etching rate of $SiO_2$. By continuously monitoring the S 2*p* signal, it started to appear after a nominal etching depth of 15 nm, which is comparable to the thickness of the Au electrode.

Next, we traced the depth profiles for each element from an etching depth of 25 nm onwards as shown in Fig. 4. The pristine capacitor exhibits a uniform distribution of all four constituent elements for the etching depth from 25 to 200 nm (Fig. 4a–d). Besides, almost no oxygen element was detected as expected (Fig. 4i). In stark contrast, the chemical composition of the fatigued capacitor varies strongly along the sample depth (Fig. 4e–h), which is accompanied by the massive oxygen diffusion into the interior of the crystal (Fig. 4j). By integrating the peak areas, we obtained the relative atomic concentration of each element as a function of the etching depth.

Apparently, the elemental distribution of the pristine capacitor is homogeneous from surface to bulk interior with negligible oxygen amount (Fig. 4k). The fatigued capacitor, however, contains substantial oxygen concentrations, which decrease gradually from ≈40% at the surface to ≈15% at 150 nm depth (Fig. 4l). Correspondingly, the relative concentrations of In and S are suppressed in the surface layer (depth <150 nm), but Cu and P elements exhibit almost constant amount within this region. However, the surface Cu concentration is significantly lower than that of the bulk interior. The surface Cu loss can be explained by the accumulation of Cu at the edge of the electrode, as revealed by the aforementioned EDS analyses (Fig. 2).

Regarding the element valence states, we didn't observe significant peak shifts or additional peaks in the XPS depth-profiling spectra of Cu 2*p*, In 3*d* and S 2*p* (Figs. S12 and S13). In contrast, the P 2*p* spectra contains the characteristic peak corresponding to the P-$O_x$ bond, whose intensity gradually diminishes in layers far from the surface. In the meantime, the peak from P-S gradually grows up. This finding is in accordance with the oxygen concentration along the depth, suggesting the oxidation of phosphorus in the surface layer of CIPS. However, we cannot exclude possible oxidation of Cu on the surface, because the binding energies of $Cu^I$-O and $Cu^I$-S bonds are very similar to each other.

**Vibrational and structural properties of fatigued CIPS**
As pointed out in previous section, the Raman spectrum after peeling off the surface layer is identical to that of a pristine CIPS (Fig. 3b). This is because in standard Raman measurements, the electric field of incident electromagnetic wave only probes the in-plane vibrational modes of the CIPS crystal. To provide more structural and bonding information along the out-of-plane direction, we cut a fatigued

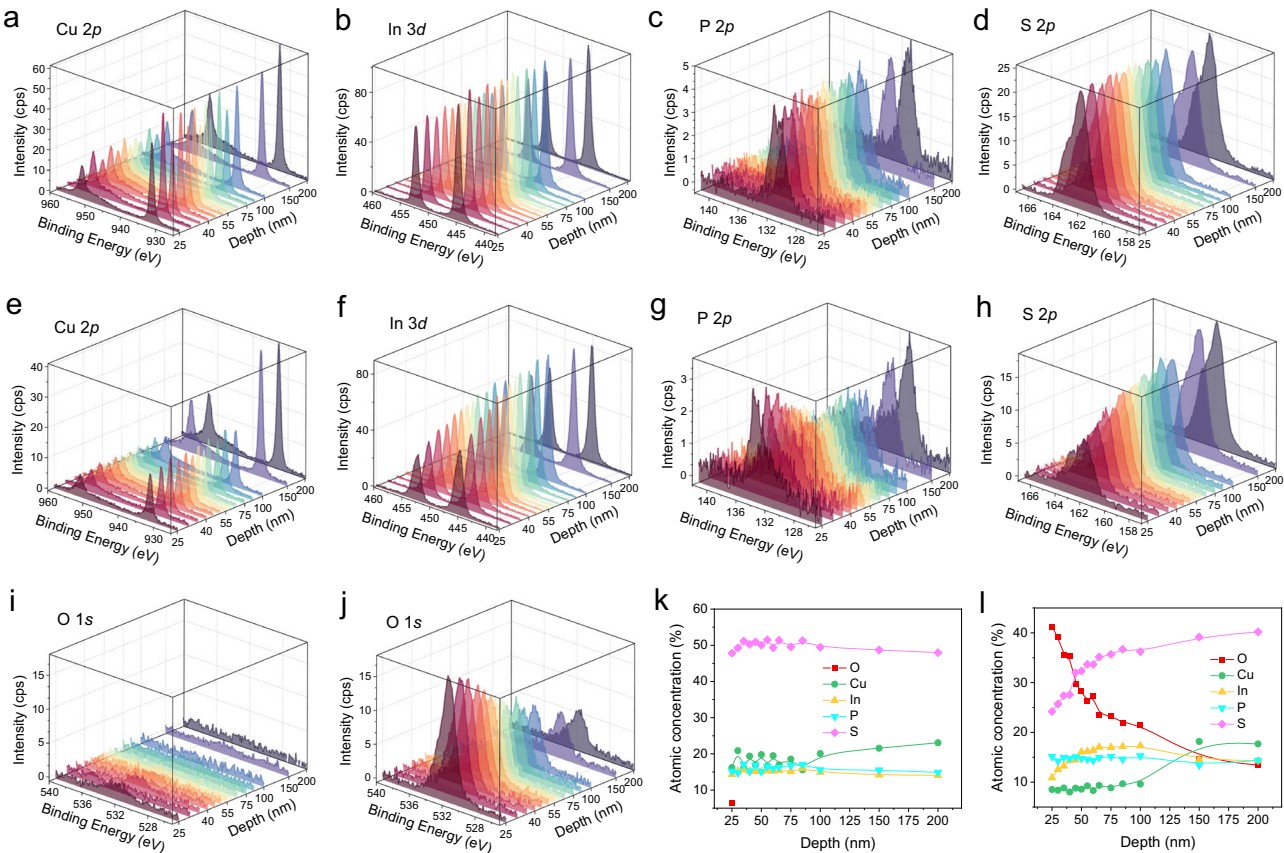

**Fig. 4 | Depth-resolved XPS study of pristine and fatigued CIPS capacitors.**
Depth-profiling of the (**a**) Cu 2*p*, (**b**) In 3*d*, (**c**) P 2*p*, (**d**) S 2*p* and (**i**) O 1*s* lines of a pristine CIPS capacitor from 25 to 200 nm. Depth-profiling of the (**e**) Cu 2*p*, (**f**) In 3*d*, (**g**) P 2*p*, (**h**) S 2*p* and (**j**) O 1*s* lines of a fatigued CIPS capacitor from 25 to 200 nm. Depth profiles of atomic concentrations for (**k**) pristine and (**l**) fatigued capacitors.

capacitor from middle and polished its cross-sectional surface for Raman study (Fig. 5a). Unlike the results of in-plane Raman spectra, the relative Raman intensity of different peaks display dramatic changes between the pristine and fatigue regions (Fig. 5b). Particularly, we selected four representative peaks for detailed analyses, namely, $P_1$ mode (~160 cm$^{-1}$) that can be assigned to the rocking motions of the $P_2S_6$ unit, $P_2$ (~240 cm$^{-1}$) and $P_3$ (~305 cm$^{-1}$) modes involving symmetric and asymmetric bending of the $PS_3$ bonds respectively, and $P_4$ (~375 cm$^{-1}$) mode driven by the symmetric P-P dimer stretching (Supplementary Movie 3−6). By performing 2D Raman mapping and plotting respective peak parameters (integrated intensity, peak position and full width at half maximum, FWHM), it helps us gain a direct comparison between the pristine and fatigued regions of the CIPS crystal (Fig. 5c−f). Apparently, $P_4$ mode is least affected, despite slight reduction in peak intensity. Since the P-P stretching is a purely intra-layer mode with in-plane S atom vibrations, the result reveals that the $P_2S_6$ structural unit remains almost intact during the electrical cycling. In contrast, the other three modes exhibit strong intensity suppression in the fatigued region. The common feature of $P_1$ – $P_3$ modes is the prominent out-of-plane motions of the S atoms. This then implies that the structural order along the interlayer direction is severely deteriorated, as also evidenced by the increase of the $P_1$ line width. Further-more, we found that the peak position of $P_1$ mode redshifted in the fatigued region, while that of $P_2$ mode blueshifted compared to the pristine state. Although both modes involve out-of-plane motions of S atoms, the specific S4 atom that interacts with the Cu in adjacent layer vibrates within the plane for $P_1$ mode. However, in $P_2$ mode, all S atoms show large out-of-plane displacements, resulting in sizable modulation of layer thickness (Supplementary Movie 3−6). From these findings, it

can be inferred that in the fatigued sample, the Cu ion possibly goes into the vdW gap, which adds an extra interlayer interaction force to the out-of-plane oscillator spring, leading to the stiffening of $P_2$ vibration mode. This hypothesis also explains the increased lattice disorder along interlayer direction, which causes an overall intensity reduction of out-of-plane vibrational modes.

Micro-X-ray diffraction (µXRD) may provide more clues to the structural changes. To this end, the incident X-ray was focused to a beam size of 50 µm to detect the structural changes due to fatigue test within an electrode. As shown in Fig. S14, the 2D detector allows us to record the conventional 2θ-θ scan as well as the rocking curve along χ direction. In the 2θ-θ pattern, the peak intensities of the fatigued capacitor are reduced compared to the pristine one, while in the χ scan, the pattern of the fatigued sample includes a sharp central peak and a diffused background. These results are in good agreement with the worsening of the lattice order as well as enhanced mosaicity of the basal plane due to the rippling of the vdW layers as evidenced by cross-sectional Raman spectra.

Moreover, quantitative PFM was employed to shed light on the domain evolution and local piezoresponse in the fatigued capacitor. Pristine and fatigued CIPS capacitors were firstly poled into one polarization state, followed by the removal of top electrode to facil-itate the measurement. Two different measurement modes were used for direct comparison, so as to confirm the reliability of our quanti-tative PFM results (Fig. S15). Following the established measurement protocol, the local piezoresponse and ferroelectric domains of the pristine and fatigued capacitors were quantitatively imaged as shown in Fig. S16. The pristine sample displays a homogeneously polarized state as confirmed by the uniform piezoresponse amplitude and phase

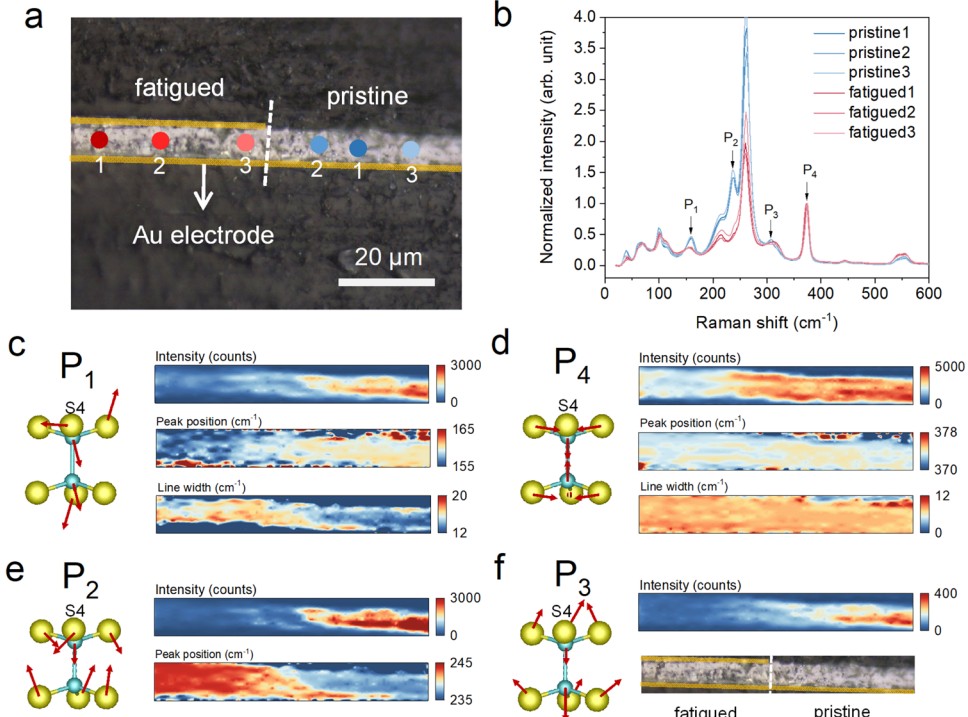

**Fig. 5 | Vibrational fingerprints of fatigued CIPS. a** Optical image of the cross-sectional CIPS capacitor with both fatigued and pristine regions. **b** Raman spectra collected at the repective spots denoted in (**a**). The intensity of the spectra is normalized to $P_4$ peak. **c**–**f** Left panel: schematic atomic motions of the related phonon modes labeled in (**b**). Right panel: corrsponding Raman spectra maps of respective phonon modes, presenting peak intensity, position and line width for $P_1$ and $P_4$ modes; peak intensity and position for $P_2$ mode; peak intensity and mapping area for $P_3$ mode.

images (Fig. S16f, 16i). In comparison, the fatigued capacitor with enhanced polarization contains regions with suppressed piezo-response (Fig. S16g, 16h, 16j, 16k). Besides, scattered small domains with opposite phase signal can be observed. There are two possible explanations for these small domains. It could be the frozen domains, which are not switchable during the pre-poling process. Or, it can be interpreted as the high-polarization phase with positive $d_{33}$, and consequently opposite phase signal23. With only the PFM results, it is not possible for us to distinguish these two scenarios. However, our micro-XRD (Fig. S14) results completely ruled out the existence of the high-polarization phase in the fatigued capacitor.

### Ionic conductivity of fatigued CIPS

The above results strongly hint the intimate link between Cu ion migration and the fatigue behaviors. Dielectric spectroscopy was used to evaluate the polarization dynamics of the fatigued CIPS in frequency domain. Owing to its displacive instability, Cu ion is the source of both the fast dipolar polarization and slow ionic migration process, whose relaxation dynamics could be reflected in the high frequency and low frequency parts in dielectric spectrum, respectively. The real and imaginary parts of the dielectric permittivity of the CIPS capacitors at different fatigue stages are plotted as a function of temperature and frequency in Fig. 6a, b. The real dielectric permittivity of the pristine CIPS shows a typical anomaly at the ferroelectric-paraelectric Curie temperature ($T_C$) of ~315 K, whereas the rise of the low-frequency imaginary part above ~ 250–260 K corresponds to the onset of thermally-activated ionic conduction[27,33]. With increasing fatigue cycles, additional contribution starts to appear and gradually intensifies in the real dielectric permittivity as indicated by the arrows. Meanwhile, the dielectric peak at $T_C$ is strongly suppressed. This behavior is somewhat analogous to the case of relaxor ferroelectrics, which demonstrate similar frequency-dependent dielectric enhancement due to the release of the frozen electric dipoles[34]. This finding is

consistent with the inhomogeneous piezoresponse and pinned domains revealed in PFM images (Fig. S15). For the imaginary permittivity, increased loss and frequency dispersion is observed in the fatigued sample at low temperature, while the ionic conduction of the fatigued sample exhibits comparable onset temperature as the pristine one, yet with much higher conductivity. To quantitatively assess the capability for ionic migration, conductivity at DC limit was first derived by fitting the AC conductivity to Jonscher's power law[35],

$$\sigma = \sigma_{DC} + A(2\pi f)^s \qquad (1)$$

where A is the pre-exponential constant, and $s$ is the power law exponent with $0 < s < 1$ (Fig. S17). The extracted temperature-dependent DC conductivity of the pristine and fatigued samples are presented in Arrhenius plot (Fig. 6c), the slope of which stands for the associated activation energy of the ionic conduction process. Apparently, the fatigued CIPS possesses a much larger conductivity and lower activation energy compared to that of the pristine sample, which is consistent with the observation in hysteresis loop measurements.

## Discussion

By bringing together the aforementioned results and evidence, we can conclude that the polarization fatigue characteristics of CIPS is closely related to its ionically mobile nature at room temperature. The displacive instability of the monovalent Cu[+] ion results in its off-center ordering and spontaneous polarization in CIPS. In the meantime, the enhanced thermal motion of the Cu[+] ion with ascending temperature eventually unleashes its intralayer or interlayer site hopping, giving rise to finite ionic conductivity. A potential Cu[+] migration path is schematically depicted in Fig. 7a in accordance with its metastable interlayer site[23–25]. Under repetitive electric cycling, it is expected that some Cu[+] ions could be occasionally trapped in the vdW gap and even extracted

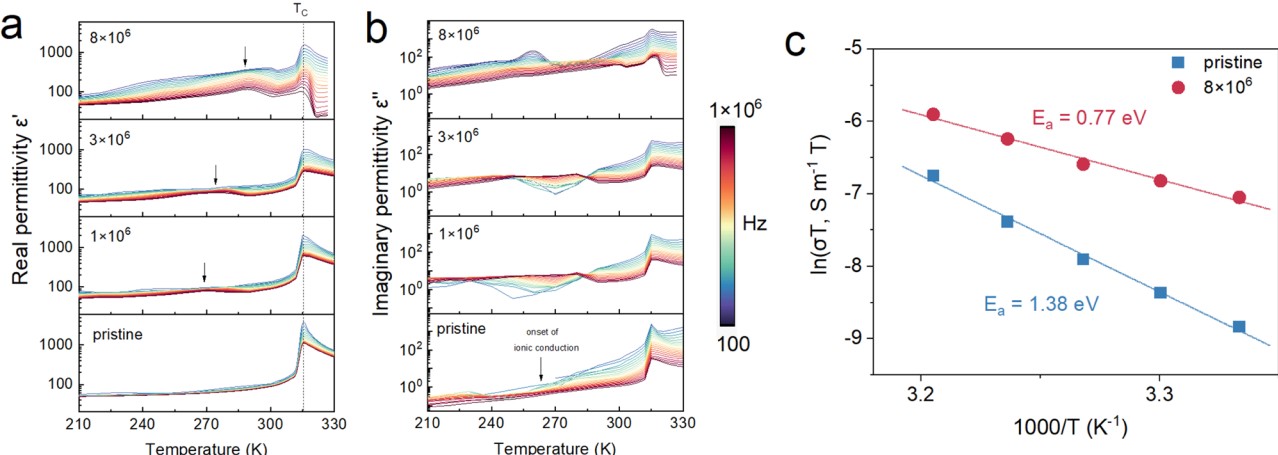

**Fig. 6 | Dielectric spectrum and ionic conductivity. a** Real and (**b**) imaginary part of the dielectric permittivity as a function of temperature for CIPS capacitors with different switching cycles. **c** Arrhenius plots of the DC conductivity for pristine and fatigued CIPS samples.

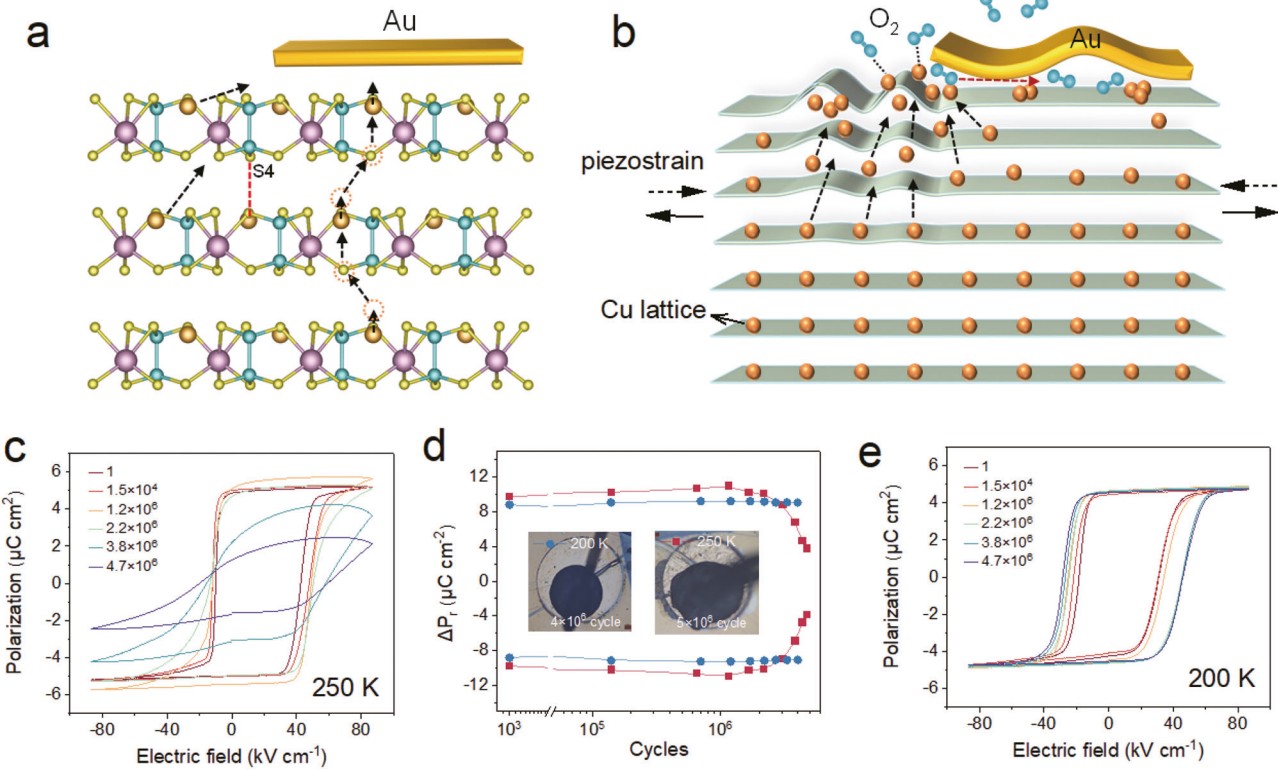

**Fig. 7 | Microscopic mechanism for the fatigue characteristics of CIPS. a** Possible Cu migration path in the crystal lattice of CIPS. **b** Schematic drawing depicting the microscopic process related to the fatigue characteristics of CIPS. **c** Sequential polarization hysteresis loops of CIPS capacitor fatigued at 250 K. **d** Switchable remanent polarization as a function of the switching cycles measured at 200 and 250 K under an electric field of $4E_C$. **e** Sequential polarization hysteresis loops of CIPS capacitor fatigued at 200 K.

out of the lattice because of the coexistence of polarization switching and ionic conduction, as depicted in Fig. 7b. Subsequently, accumulated Cu$^+$ could be electrochemically oxidized by the oxygen in the air as evidenced by the enrichment of oxygen and Cu at the electrode edge in EDS mapping. This effect is most prominent at the CIPS-Au-air interface, where the electric field is strongly concentrated. Although bipolar field was used in the study, the electrochemical process is irreversible. As a result, Cu atoms are gradually depleted from the crystal lattice, which leads to an overall decrease of Cu amount in the

surface layer within the capacitor as verified by the depth-profiling XPS (Fig. 4). This effect causes empty sulphur octahedra to rearrange into $S_8$ structures, as detected by the Raman spectrum. Additionally, local strains due to ion concentration variation (direct Vegard effect)[36] and/ or electric polarization gradient (converse flexoelectric effect)[37–39] may strongly disturb the crystal lattice order perpendicular to the vdW layers due to random Cu$^+$ hopping motions, which is confirmed by the cross-sectional Raman and μXRD results. Dipole-glass like relaxation process and enhanced DC ionic conductivity as derived from the

dielectric spectra further corroborates the microscopic ionic dynamics during the fatigue process.

The anomalous morphological evolution, such as bubble-like protrusion and surface wrinkling, is closely related to the cooperation of oxidation and switching-induced piezostrain during electric cycling. Our time-lapse video clearly demonstrated that the darkening of the electrode edge (an indication of the oxidation) and the protrusion outgrowth initiate concurrently (Supplementary Movie 1). A possible explanation is that the oxidation breaks the integrity of the crystal lattice and the electrode-sample interface at the edge, enabling the transport of air into the interior of the capacitor. Meanwhile, piezo-strains generated by the periodic bipolar voltage will drive the contraction and expansion of the crystal lattice within the capacitor, which facilitates the partial delamination between the electrode and the crystal surface, resulting in repetitive inflation and deflation of the bubble-like protrusions. Similarly, the oxidation in the surrounding region also promotes the corrugation of the crystal layers in the surrounding surface through the ripplocation effect due to the weak interlayer vdW interaction[40,41].

Regarding to the fatigue behaviors, polarization enhancement is generally observed before the final breakdown of the device due to the devastating leakage. Such anomalous polarization enhancement is different from commonly reported wake-up effect in hafnium oxide and other oxide ferroelectrics[42–46], as the increased polarization value is way beyond the intrinsic polarization of CIPS. Our structural and PFM analyses have ruled out the possible phase transformation to the high-polarization phase that contributes to the polarization enhancement. However, the strongly increased electrical conductivity in the rippled area around the electrode vicinity make us to speculate that the polarization enhancement could originate from an increase of actual switching area. To testify our guess, we isolated the conductive area from the central capacitor by manually scratching the top electrode using micro-manipulator probe station (Fig. S18). Indeed, the measured polarization of the re-defined electrode area decreased back to the nominal value of CIPS. In fact, by normalizing the measured polarization charges to its actual switching areas, similar polarization value is observed for different cases. The result unambiguously suggests the enhancement of the polarization originated from an increase in the actual electrode area due to the formation of the conductive surface layer. Electric cycling under low field and frequency seems to promote the uniform formation of the oxidized conductive layer, possibly due to a longer exposure time to air. In contrast, high field will produce larger transient strain during polarization switching, facilitating the ripplocation effect and resulting in large-scale interlayer delamination as seen in the optical images. Besides, large cycling field may inevitably lead to long-range ionic migration and random hopping of the $Cu^+$ ions, followed by the formation of the leakage path and loss of polarization due to Cu extraction from lattice. From the chemical degradation sense, the fatigue behavior of CIPS shares some similarities with several previous cases, such as local phase decomposition in oxide ferroelectrics[12] and organic ferroelectrics[32], as well as metal electrode cation permeating into the oxide ferroelectrics[47].

To corroborate the synergetic effect of ionic conduction and surface oxidation in the fatigue process, we conducted additional fatigue tests at low temperature in air to suppress the ionic conduction effect (Fig. 7c–e). At 250 K, the CIPS capacitor shows similar fatigue behavior as those tested at room temperature, because the ion is still mobile at this temperature. However, the morphologic degradation is greatly alleviated. Further decreasing the temperature to 200 K, we found basically no decay of switchable polarization up to $4 \times 10^6$ cycles with an electric field as high as ~$4E_C$. The temperature-dependent results clearly suggest that the polarization fatigue issue can be resolved once the ionic conduction is turned off. Moreover, fatigue tests at room temperature yet under vacuum (Pressure <$10^{-3}$ Pa) were also performed to check the oxidation effect (Fig. S19).

Apparently, with no oxygen present, the CIPS capacitor didn't show significant morphological deterioration and polarization reduction up to $10^7$ cycles.

In summary, we report the polarization fatigue characteristics of the vdW ferroelectric CIPS with finite ionic conductivity at room temperature. Unlike conventional ferroelectrics, the relatively small difference between the energy barriers for intralayer polarization reversal and interlayer ionic migration makes them both active under repetitive electric cycling, introducing unconventional behaviors in the fatigue process. The drastic changes in the morphology including the severe surface ripples and bubble-like protrusions can be attributed to the electrochemical extraction of the Cu atoms from the lattice and strain-induced delamination and corrugation of the vdW layers. The field-assisted oxidation mainly at the electrode edge and surrounding vicinity results in the formation of a conductive layer that expands the effective switching area, which explains the anomalous increment of switchable remanent polarization at the early stage of the fatigue process. The eventual polarization reduction can be attributed to the crystal phase degradation and leakage path formation due to the substantial loss of Cu ions in the lattice. The intriguing fatigue characteristics reported here greatly expand the understanding of the switching durability in unconventional classes of ferroelectrics, and provide insights into the strategy for relieving the fatigue effect by suppressing ionic activity or undesirable oxidation in practical applications.

## Methods

### Materials synthesis and device fabrication

$CuInP_2S_6$ single crystals were synthesized by chemical vapor transport method using stoichiometric mixture of the constituent element powders as the precursors and iodine as the transport agent. The reaction was carried out in an evacuated quartz ampoule located in a two-zone furnace (750/700 °C) for a reaction time of several days. High-quality and regular-shape single crystals were carefully picked out from the raw reaction products by optical microscopy inspection. The composition stoichiometry and phase purity were checked by energy dispersive X-ray spectroscopy (EDS) and Raman spectroscopy. To fabricate the parallel-plate ferroelectric capacitors, the bottom face of the crystals first coated with Au electrode by sputtering. The top electrodes were defined by shadow masks to form circular or square pads with a dimension of 200–300 μm. The detailed samples/devices information and involved experiments are included in the Supplementary Information (Table S1). The exact electrode areas were precisely calculated using calibrated optical images.

### Electrical measurements

The polarization switching characteristics such as standard polarization hysteresis loop, remanent hysteresis loop and PUND measurement were conducted using a commercial ferroelectric tester (Radiant Technologies, Multiferroic II). The fatigue tests were performed using square bipolar pulse with different amplitudes and frequencies, and the switchable remanent polarization was measured using PUND method. Complex dielectric data were collected by a commercial LCR meter (Agilent, 4284 A). Temperature dependent electrical measurements were carried out in a cryogenic micromanipulator probe station equipped with a heating stage (Instec, HCS621G).

### Chemical and structural characterizations

The EDS spectra and spatial mapping were collected by the energy dispersive spectrometer (Bruker, QUANTAX) equipped on a commercial field-emission scanning electron microscope (Hitachi, Regulus 8100). A 20 kV accelerating voltage was used to ensure the excitation of the sufficient X-ray lines for the elements. Standardless quantitative analysis of the sample composition is achieved by calculating the ratio of peak intensities to determine the relative abundance of the element.

For the elemental mapping, a long acquisition time of at least 5 min were used to ensure the measurement accuracy and reproducibility.

Raman spectra were measured by a confocal Raman microscope with a 532 nm laser and a grating of 2400 gr mm$^{-1}$, producing a spectral resolution better than 1 cm$^{-1}$ (Horiba, XploRA Plus). In spatial Raman mapping, the fiber-coupled laser was focused by a long working distance objective (NA = 0.5, WD = 10.6 mm) into a Gaussian beam with a diameter less than 1 μm. The raw data was processed and analyzed by LabSpec6 software.

Depth-profiling XPS was carried out on a micro-focus X-ray photoelectron spectrometer (Thermo Scientific, Nexsa G2) with monochromated Al K$_\alpha$ source (1486.6 eV) and a beam diameter of 50 μm. Depth profiling was performed by etching the surface using Ar$^+$ ion beam for different durations. The nominal etching thickness is estimated based on the etching rate of SiO$_2$.

μXRD was performed on a commercial X-ray diffractometer (D8 Discover, Bruker) with an area detector. A Gobel mirror parallel optics system was used to remove the Cu K$_\beta$ radiation. A pinhole collimator was used to focus the Cu K$_\alpha$ beam down to a diameter of 50 μm onto the sample surface.

### Multi-functional scanning probe microscopy

All scanning probe microscopic imaging were performed on a commercial atomic force microscopy (Oxford Instrument, MFP-3D origin + ). The ambient PFM studies were carried out using both dual AC resonance tracking (DART) method with a Ti/Pt-coated tip (OMCL-AC240TM, resonant frequency ≈70 kHz, spring constant ≈2 N/m) and under non-resonant mode with a hard Pt/Ir-coated tip (Nanoworld-NCHPt, resonant frequency ≈320 kHz, spring constant ≈42 N/m). The inverse optical lever sensitivity (InvOLS) was calibrated using combined Sader method and thermal noise method to obtain quantitative piezoelectric displacements in PFM images. For DART-PFM, the PFM data are fitted by simple harmonic oscillator (SHO) model to obtain effective piezoelectric response off resonance. SKPM was carried out using a two-path method, where the topographic image was acquired in the first path and in the second path, surface potential can be extracted by nulling the electrostatically induced oscillation of the cantilever. In CAFM, a DC bias of 5 V was applied between the probe and bottom electrode to measure the spatial distribution of the current. In both SKPM and CAFM, the Ti/Pt-coated tip (OMCL-AC240TM) was used.

## Data availability

The data that support the findings of this paper are available from the corresponding authors upon request.

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

## Acknowledgements
L.Y. and L.F. acknowledge the support by the National Natural Science Foundation of China (No. 12074278), the Natural Science Foundation of the Jiangsu Higher Education Institution of China (20KJA140001), and the Priority Academic Program Development (PAPD) of Jiangsu Higher Education Institutions. L.Y. also acknowledges the support from Suzhou Science and Technology Bureau (ZXL2022514) and Jiangsu Specially-Appointed Professors Program. The authors are grateful for the technical support of Nano-X from Suzhou Institute of Nano-Tech and Nano-Bionics, Chinese Academy of Sciences (SINANO).

## Author contributions
L.Y., Y.W. and L.F. designed and supervised the project. S.W. and Z.Z. synthesized the single crystals. Z.Z. and S.W. fabricated the devices and conducted electrical tests and ferroelectric fatigue measurements. Z.Z. and S.W. conducted EDS and Raman measurements. Y.H., Q.L., Z.L. carried out scanning probe microscopy measurements. S.W. and L.K. performed XPS measurement and analysis. K.O. and H.F. conducted μXRD tests. L.Y. and Z.Z. co-wrote the manuscript. L.F., Y.W., J.X., Z.F., Q.Y., R.T., X.S., and F.Z. commented and gave suggestions on the manuscript.

## Competing interests
The authors declare no competing interests.
