## [Peer Review File · Nature Communications]

REVIEWER COMMENTS

Reviewer #1 (Remarks to the Author):

The long-term operation of ferroelectric-based devices largely depends on polarization fatigue. Van der Waals ferroelectric materials have attracted widespread interest due to the continuing need for device scaling and enhanced functionality. The topic of polarization fatigue in the van der Waals ferroelectric material CuInP2S6 is timely. However, I find that observations and discussions in this manuscript do not well support its conclusions. Therefore, in my opinion, this manuscript needs more work before it can be considered by Nature Communications.

(1) The basic information of the material is not well provided in the manuscript, such as whether it is a single crystal (if not, domain walls can greatly affect fatigue performance). In particular, the thickness of the material is only mentioned in the Method as 10-20 μm . Each sample should be accurately measured in thickness, and the thickness should be used to normalize the voltage to the electric field. Also, the magnitude of the electric field should be included in the discussion to validate the proposed model.

(2) Due to the huge impact of van der Waals ferroelectricity, I strongly suggest that one of the measurements (more importantly, polarization hysteresis loops) should be done on a few-layer device. This will also be used to detect any differences between bulk samples and few-layer samples.

(3) The "strongly" suggested conclusion that Cu ions are extracted from the lattice is not well supported by the experiments and discussions.

(i) The increase in Cu concentration in the fatigued region is not convincingly demonstrated by EDS. With such a small value difference, EDS is not sensitive enough.

(ii) What is the source of the increased Cu atoms? If it is because the Cu ions extracted from the lattice of the inner materials are oxidized and fixed on the surface near the electrodes, and EDS also detects elemental information more from the surface, I strongly recommend that authors perform EDS mapping of their cross-sectional samples.

(iii) More characterizations, such as XPS and TEM, are needed to probe element valence states and lattice structure information of fatigued samples.

(4) The anomalous increase in polarization reported in the manuscript, especially in relation to polarization fatigue, should be clearly understood before resubmitting. It is unacceptable to leave this as an unsolved issue, as it could be very relevant.

(5) The scale bar should be indicated in Fig. 3a.

Reviewer #2 (Remarks to the Author):

Zhou et al reported the observation of unusual polarization fatigue behaviours in vdW layered CuInP2S6 with the finite ionic conductivity at room temperature. They found the unique morphological and polarization evolutions under repetitive electric cycles origins from the strong intertwinement of the short-range polarization switching and long-range ionic movement in conjunction with the vdW layered structure. The microscopic mechanisms for the anomalous polarization enhancement and polarization degradation was discovered. This is an interesting work about the polarization fatigue behaviours in ionically-active vdW ferroelectrics. It can be published after the following issues are address:

1.The author claims that Cu ion migration plays an important role in polarization aging. However,

there is no apparent evidence of Cu enrichment in the EDS mapping characterization presented in Figure S5, S6.

2.The study focuses on the bulk single crystal. What about thin flakes? is there thickness dependence on the fatigue behavior?

3.In Fig. 4a, there is an extra shoulder peak by the side of the main peak. What is the origin for this extra peak?

4.The exact PFM measurement conditions for three specimens In Fig. 4 should be provided. In order to directly compare the magnitude of amplitude signals, there should be an identical reference region included during the PFM measurement, since the measuring conditions and cantilever tunings are different. Moreover, in Fig. 4e and f, where amplitude is characterized using PFM, the amplitude contrast (bright/dark) does not necessarily indicate the coexistence of polarized phases, such as low and high polarization states. Electrostatic forces can significantly influence the PFM results, often causing distinct amplitude contrasts between the upper and lower polarized phases. It is recommended that the author carefully address and eliminate these factors. I recommend the authors follow the recent recipe of "electrostatic blind spot" (<https://arxiv.org/pdf/2112.09665.pdf>). Can the author provide a zoom-in scanning map in the bright and dark amplitude regions?

5.Considering the potential coexistence of low and high polarization phases with different d_{33} values, it would be more interesting and compelling if the fatigue of d_{33} and polarization can be investigated.

6.Compared to previous studies (ACS Applied Materials & Interfaces, 2022, 14(2): 3018-3026), the author should emphasize the importance and significance of their work more prominently.

7.Some details in the article require careful check. For example, the scale bar length is not indicated in Fig.3. the 100 microns scale bar in Fig. 1 is too small. Furthermore, the experimental methods section lacks sufficient details. It is recommended that the author provide a comprehensive supplementation of these missing details.

8.The mechanism of induced rippling is illustrated as Vegard effect and ripplation effect. Can the authors comment on this if the converse flexoelectric effect is also contributing?

9.In oxide ferroelectrics, one effective way to relieve the polarization fatigue is to use oxide electrodes instead of metal electrodes, so as to suppress the formation of oxygen vacancies. I wonder for CIPS whether we can employ similar strategy. Can the author comment on this issue?

Response to Reviewer #1

REVIEWER COMMENTS

Reviewer #1 (Remarks to the Author):

The long-term operation of ferroelectric-based devices largely depends on polarization fatigue. Van der Waals ferroelectric materials have attracted widespread interest due to the continuing need for device scaling and enhanced functionality. The topic of polarization fatigue in the van der Waals ferroelectric material CuInP_2S_6 is timely. However, I find that observations and discussions in this manuscript do not well support its conclusions. Therefore, in my opinion, this manuscript needs more work before it can be considered by Nature Communications.

Response: We thank the reviewer for pointing out the flaws in the manuscript and providing valuable suggestions. We have performed additional experiments and collected new results to further clarify the underlying fatigue mechanism. We believe the manuscript is now greatly improved and addresses all the questions and suggestions by the reviewer. Please see the point-by-point responses below for details.

(1) The basic information of the material is not well provided in the manuscript, such as whether it is a single crystal (if not, domain walls can greatly affect fatigue performance). In particular, the thickness of the material is only mentioned in the Method as 10-20 μm . Each sample should be accurately measured in thickness, and the thickness should be used to normalize the voltage to the electric field. Also, the magnitude of the electric field should be included in the discussion to validate the proposed model.

Response: We apologize for the incomplete information in Method. All the samples measured in the study are single crystals. Following the reviewer's suggestion, we have added a complete description of sample details in the supporting information, including the sample thickness, electrode size, coercive field, together with characterization methods involved. The voltage values of polarization hysteresis loops in the manuscript figures are also normalized to electric field. All the crystals involved in the study show the coercive field in the range of 25 – 29 kV/cm.

(2) Due to the huge impact of van der Waals ferroelectricity, I strongly suggest that one of the measurements (more importantly, polarization hysteresis loops) should be done on a few-layer

device. This will also be used to detect any differences between bulk samples and few-layer samples.

Response: We thank the reviewer for this good suggestion. There are currently two difficulties that prevent us from measuring few-layer samples. First, the capacitor becomes too leaky for polarization hysteresis loop measurement below 100 nm, and secondly, the area of the exfoliated few-layer flake is usually too small (less than $10 \times 10 \mu\text{m}^2$) for a reliable measurement of the switched polarization. Nevertheless, we still carried out fatigue study on a 172-nm-thick thin flake of CIPS as shown in **Fig. R1**. The fresh capacitor shows well-defined polarization hysteresis loop with a remanent polarization of $\approx 3.5 \mu\text{C cm}^{-2}$. With increasing cycles, the remanent polarization of the thin flake displays similar incremental behavior to that of the bulk sample, until a sudden breakdown close to 10^6 cycles. We have included the result in the supporting information.

Fig. R1. (a) Optical image of the cross-bar capacitor of a CIPS thin flake. The inset shows the AFM topographic image to determine the flake thickness. (b) Remanent hysteresis loop of the thin flake capacitor by subtracting the non-remanent contribution of the polarization. (c) Polarization hysteresis loops of the thin flake capacitor after cumulative switching cycles. (d) Remanent polarization as a function of the switched cycles.

(3) The “strongly” suggested conclusion that Cu ions are extracted from the lattice is not well supported by the experiments and discussions.

(i) The increase in Cu concentration in the fatigued region is not convincingly demonstrated by EDS. With such a small value difference, EDS is not sensitive enough.

(ii) What is the source of the increased Cu atoms? If it is because the Cu ions extracted from the lattice of the inner materials are oxidized and fixed on the surface near the electrodes, and EDS also detects elemental information more from the surface, I strongly recommend that authors perform EDS mapping of their cross-sectional samples.

(iii) More characterizations, such as XPS and TEM, are needed to probe element valence states and lattice structure information of fatigued samples.

Response: We totally agree with the reviewer that the spatial resolution of EDS ($\approx 1 \mu\text{m}$) is much larger than the beam size itself ($\approx 10 \text{ nm}$) due to the large electron interaction volume. In fact, we have tried EDS mapping of cross-sectional samples. However, due to its poor spatial resolution, we didn't observe any noticeable element redistribution in the fatigued sample.

Following the reviewer's suggestion, we then conducted depth-profiling XPS on both pristine and fatigued capacitors. The sample details are shown in **Fig. R2**. The probed areas were located at the center of the capacitors with an X-ray beam size of $\approx 50 \mu\text{m}$ in diameter. The depth profiling was performed by sequential Ar^+ ion etching, in which the nominal thickness is estimated based on the etching rate of SiO_2 . By continuously monitoring the S 2p signal, it started to appear after a nominal etching depth of 15 nm, which is comparable to the thickness of the Au electrode.

Fig. R2. Fatigued properties of the sample for XPS study. (a) Optical image of the CIPS crystal with pristine and fatigued capacitors. The color dots denote the probed areas (diameter $\approx 50 \mu\text{m}$). (b) S 2p

spectra at nominal etched depths. (c) Polarization hysteresis loops of the capacitor after different switching cycles. (d) Switchable remanent polarization as a function of the electrical cycles.

Fig. R3. Depth-resolved XPS study of pristine and fatigued CIPS capacitors. (a-d, i) Depth-profiling of the (a) Cu 2p, (b) In 3d, (c) P 2p, (d) S 2p and (i) O 1s lines of a pristine CIPS capacitor from 25 – 200 nm. (e-h, j) Depth-profiling of the (e) Cu 2p, (f) In 3d, (g) P 2p, (h) S 2p and (j) O 1s lines of a fatigued CIPS capacitor from 25 – 200 nm. Depth profiles of relative atomic concentrations for (k) pristine and (l) fatigued capacitors.

Next, we traced the depth profiles for each element from an etching depth of 25 nm onwards as shown in **Fig. R3**. The pristine capacitor possessed a uniform distribution of all four constituent elements for the etching depth from 25 to 200 nm (**Fig. R3a-d**). Besides, almost no oxygen element was detected as expected (**Fig. R3i**). In stark contrast, the chemical composition of the fatigued capacitor varies strongly along the sample depth (**Fig. R3e-3h**), which is accompanied by the massive oxygen diffusion into the interior of the crystal (**Fig. R3j**). By integrating the peak areas, we obtained the relative atomic concentration of each element as a function of the etching depth. Apparently, the elemental distribution of the pristine capacitor is homogeneous from surface to the bulk interior with negligible oxygen amount (**Fig. R3k**). The fatigued sample, however, contains substantial oxygen concentrations, which decrease gradually from 40 % at the surface to ≈ 15 % at 150 nm depth (**Fig. R3l**).

Correspondingly, the relative concentrations of In and S are suppressed in the surface layer (depth < 150 nm), but Cu and P exhibit almost constant amount in this region. However, the surface Cu concentration is significantly lower than that of the bulk interior. The surface Cu loss can be explained by the accumulation of Cu at the edge of the electrode, as revealed by our detailed EDS analysis (Fig. R4). As shown in Fig. R4b, with the increase of O concentration at the electrode edge, the relative concentration line profiles of In, P and S displayed corresponding dips at the same location as expected, because the total atomic concentration sums up to 100 %. However, this case was not observed for Cu, suggesting a relative enrichment of Cu at the electrode edge. This result supports the fact of mass transfer of Cu from the capacitor interior (only the surface layer) towards its edge due to continuous oxidation during electric cycling.

Fig. R4. (a) The SEM image of a pristine and a fatigued CIPS capacitor. (b) EDS spectra averaged over different areas marked by the corresponding color boxes in (a). The dash box highlights the stark

difference at O K α line. (c) The corresponding oxygen distribution map of (a). (d) EDS line profiles of different elements along the dash line shown in each map. (e) Cu, (f) P, (g) In, and (h) S distribution map of (a).

Regarding the element valence states, we didn't observe significant peak shifts or additional peaks in the XPS depth-profiling spectra of Cu 2p, In 3d and S 2p (**Fig. R5**). In contrast, the P 2p spectra contains the characteristic peak corresponding to the P-O $_x$ bond, whose intensity gradually diminishes in layers far from the surface. In the meantime, the peak attributed to P-S gradually grows up. This finding is in accordance with the oxygen concentration along the depth, suggesting the oxidation of phosphorus in the surface layer of CIPS. However, we cannot exclude possible oxidation of Cu on the surface, because the binding energies of Cu^I-O and Cu^I-S bonds are very similar to each other.

Fig. R5. (a) Cu 2p, (b) In 3d, (c) P 2p, (d) S 2p and (e) O 1s XPS spectra of a fatigued CIPS capacitor from a depth of 25 to 200 nm. The data are the same as those in Fig. R3, but presented in stacking plots.

In terms of TEM study, we found the cross-sectional sample fabrication was highly challenging and currently beyond our reach. To gain more insight from the lattice structure of fatigued sample, we have performed Micro X-ray Diffraction (μ XRD). It should be noted that our previous XRD results were obtained using conventional XRD, which has a large beam size that covers the whole sample. Hence, it may not accurately reflect the structure information of the fatigued area. Here, we focused the incident X-ray to 50 μ m to detect the structural changes due to fatigue test within an electrode. As shown in **Fig. R6**, the 2D detector allows us to record

the conventional 2θ - θ scan and rocking curve along χ direction. In the 2θ - θ pattern, the peak intensities of the fatigued capacitor are reduced compared to the pristine one, consistent with our previous result. However, this time we didn't observe noticeable peak shift induced by fatiguing. Actually, in our previous XRD results, the shift of (008) peak is tiny (from 56.472° to 56.482°), which translates to a change of only 0.0024 \AA in d spacing of (001) plane. This tiny lattice change is incompatible with the low-polarization to high-polarization phase transition, as proposed previously (Nat. Mater., 19, 43–48, 2020). Therefore, we believe the experimentally observed polarization enhancement is caused by other effect as detailed in the following response. In the χ scan, the pattern of the fatigued sample includes a sharp central peak and a diffused background, which can be attributed to the rippling of the vdW layers. This effect, however, is much weaker within the electrode than in the surroundings, as shown in our previous XRD.

Fig. R6. 2D XRD maps of (a) pristine and (b) fatigued CIPS capacitors around (006) and (008) bragg peaks. (c) 2θ - θ scan of the (006) and (008) peaks of the CIPS capacitors. The inset is the zoomed-in plot of the (008) peak. (d) χ scan of the (008) peak of the CIPS capacitors.

(4) The anomalous increase in polarization reported in the manuscript, especially in relation to polarization fatigue, should be clearly understood before resubmitting. It is unacceptable to leave this as an unsolved issue, as it could be very relevant.

Response: We appreciate the reviewer for this important comment, which motivates us to carry out more comprehensive experiments to clarify this issue, and consequently gain more insight into the fatigue behavior of CIPS.

Firstly, we have conducted multiple microscopic imaging of the fatigued capacitor with anomalous morphological changes, as shown in **Fig. R6**. Two representative regions were investigated, namely, the electrode edge and the rippled area that spreads far way, as denoted by the blue and red boxes in **Fig. R6a**. Correlated with the optical images (**Fig. R6d, j**), the AFM topographic image (**Fig. R6e**) of the rippled area is featured by hillocks with tens to hundreds of nanometers in height, whereas the surface ripples near the electrode edge are so dense that they strongly merge with each other (**Fig. R6k**). Besides, the dark area at the electrode edge shown in optical image exhibits much higher outgrowth than the rippled area. Our EDS mapping of the same locations indicated obvious oxygen accumulation in the rippled area (**Fig. R6f**), which is even more intense around the electrode edge (**Fig. R6l**).

Fig. R6. Correlated microscopic imaging of the fatigued capacitor. (a) Optical image of a fatigued CIPS capacitor. (b) Raman spectra collected at representative spots marked in (d) and (j). (c) Local current-voltage curves measured at pristine and rippled area marked in (i). Correlated microscopic

images of the areas denoted by (d-i) red, (j-n) blue and (o) purple dash box shown in (a) using multiple imaging techniques: (d, j) optical images, (e, k) AFM topographic images, (f, l) O element maps, (g, m) SKPM images, (h, n) PFM images, and (i, o) CAFM images.

For other elements, because the overwhelming contribution from the underneath bulk material in EDS measurements, it is difficult for us to discern the small composition variations on the surface ripples. Nevertheless, by enhancing the contrast of the EDS map (Fig. R7), we were able to observe the difference: there is slight Cu enrichment in the rippled area compared to the other three elements, in agreement with the result found in Fig. R4.

Fig. R7. (a) AFM topographic image and (b-f) corresponding EDS elemental maps of the rippled area of the fatigued capacitor. (b) O, (c) Cu, (d) In, (e) P, and (f) S map.

Furthermore, we also imaged the surface potential (Fig. R6g, 6m) and piezoelectric response (Fig. R6h, 6n) of these two areas using scanning Kelvin probe microscopy (SKPM) and piezoresponse force microscopy (PFM). The results indicated reduced surface potential and piezoelectricity of the rippled surface compared to those of the pristine surface. Raman spectroscopy identified characteristic peaks from cyclooctasulphur S8 rings at the electrode edge (Fig. R6b). Its signature peak at 472 cm^{-1} peak due to the symmetric S-S bond stretching is also visible in the Raman spectrum recorded at the rippled surface. The peak intensity seems to directly correlate with the oxygen concentration. As our XPS result indicates that the oxidation mainly occurs in the surface layer ($< 200\text{ nm}$ depth), after peeling off the surface layer, the Raman spectrum becomes almost identical to that of a pristine surface.

Last but not least, we employed conductive AFM (CAFM) to probe the conductivity changes caused by the potential oxidation in the rippled area and electrode edge. As the measurement on top of the Au electrode would result in huge currents that might damage the tip, we chose another dark region (marked by purple box in **Fig. R6a**) near the electrode edge for imaging. Interestingly, the regions with oxygen accumulation also possess higher electrical conductivity as evidenced by the CAFM images, and the conductivity seems to scale with the oxygen amount (**Fig. R6i, o**). The local current – voltage curves confirmed the strongly enhanced conductivity at the surface ripple than at pristine surface (**Fig. R6c**). All the above results are highly correlated, and point to the important conclusion that repetitive electric cycles cause substantial surface oxidation at the electrode edge and its surrounding areas. The oxidation results in significant changes in the morphological, structural and electronic properties. No detectable oxide peak was observed in the Raman spectra, suggesting the oxidized product is probably non-crystalline. Nevertheless, oxidized surface layer possesses enhanced electrical conductivity, which may potentially increase the electrode area and the measured polarization (as we assume the electrode area is fixed).

Fig. R8. Optical images of (a) pristine capacitor, (b) fatigued after 1.3×10^6 cycles, (c) after scratched and isolated from the surrounding conductive area, and (d) after creating a smaller isolated capacitor by scratching. (e) Measured total polarization charge hysteresis loops in the four cases of (a-d). (f) Recalculated polarization values after normalizing to the actual electrode areas.

To confirm our hypothesis, we conduct a set of experiments by scratching the top electrode using micro-manipulator probe station to re-define the actual electrode area. As shown in **Fig. R8**, we chose a fresh capacitor, measured its electrode area and polarization charge (**Fig. R8a**). Then, we cycled the capacitor using an electric field of $2E_c$ for 1.3×10^6 cycles, and re-measured the hysteresis loop. As expected, the switchable polarization charge was greatly increased. This time, we also estimated the total area of the capacitor including the surrounding oxidized area (**Fig. R8b**). Next, we scratched the sample surface using tungsten probe (10 μm diameter) to isolate the central capacitor from the conductive oxidized area (**Fig. R8c**). Since the oxidation is mainly confined in the surface layer less than 200 nm depth, we could easily scratch away the oxidized layer to expose a fresh CIPS surface, which is insulating. We then re-measured the polarization hysteresis loop of the scratched capacitor, and found its switchable polarization charge reduced back to the original value! The result unambiguously suggests the enhancement of the polarization originated from an increase in the actual electrode area due to the formation of the conductive surface layer after electric cycling. For further confirmation, we re-defined a smaller capacitor by scratching (**Fig. R8d**) and measured its switchable polarization. By normalizing the total measured charges to the actual device areas, we found the polarization values are similar for these four cases, thus confirming our argument.

Our updated micro-XRD (**Fig. R6**) and PFM (**Fig. R9**) results also excluded that the phase transformation into high-polarization state as the main source for the polarization enhancement. Although in the PFM image, we did observe some domains with small piezoresponse. However, they only accounts for less than 5 % of the total area. Furthermore, the micro-XRD result completely ruled out the existence of the high-polarization phase in the fatigued capacitor. Hence, the low-piezoresponse domains are more likely frozen domains or domains with glassy dipoles, which contribute to the dielectric relaxation in the permittivity spectrum.

Fig. R9. Optical images of the CIPS single crystal (a) before and (b) after electrode exfoliation. The fresh and fatigued capacitors were poled into single polarization state before the exfoliation. The blue and red boxes indicate the scan areas for fresh and fatigued capacitors, respectively. The white arrows denote markers for positioning. (c-e) Topographic, (f-h) PFM amplitude, and (i-k) PFM phase images of the fresh and fatigued areas as indicated by corresponding color boxes. The PFM was performed in DART mode using a PtIr-coated probe ($k \approx 2 \text{ N/m}$). The piezoresponse amplitude was fitted using SHO model to obtain quantitative values.

(5) The scale bar should be indicated in Fig. 3a.

Response: We apologize for the omission. The scale bar is $20 \mu\text{m}$. We have modified the figure accordingly.

Response to Reviewer #2

Reviewer #2 (Remarks to the Author):

Zhou et al reported the observation of unusual polarization fatigue behaviours in vdW layered CuInP_2S_6 with the finite ionic conductivity at room temperature. They found the unique morphological and polarization evolutions under repetitive electric cycles origins from the strong intertwinement of the short-range polarization switching and long-range ionic movement in conjunction with the vdW layered structure. The microscopic mechanisms for the anomalous polarization enhancement and polarization degradation was discovered. This is an interesting work about the polarization fatigue behaviours in ionically-active vdW ferroelectrics. It can be published after the following issues are address:

Response: We thank the reviewer for the positive assessment of our work. Please see the point-by-point responses below for details. Note some figures have appeared in the prior section of Response to Reviewer #1, so the numbering of the figures may not be in order in this section.

1.The author claims that Cu ion migration plays an important role in polarization aging. However, there is no apparent evidence of Cu enrichment in the EDS mapping characterization presented in Figure S5, S6.

Response: We thank the reviewer for this important question. The difficulty to identify the Cu enrichment in our sample is because the spatial resolution of SEM EDS ($\approx 1 \mu\text{m}$) is much larger than the beam size itself ($\approx 10 \text{ nm}$) due to the large electron interaction volume. As a result, even if there are Cu enrichment on the surface, the small composition variation is masked by the strong background from the bulk material. To gain more insight into this issue, we performed more detailed analysis on our EDS data (**Fig. R4**). As shown in **Fig. R4b**, with the increase of O concentration at the electrode edge, the relative concentration line profiles of In, P and S displayed corresponding dips at the same location as expected, because the total atomic concentration sums up to 100 %. However, this case was not observed for Cu, suggesting a relative enrichment of Cu at the electrode edge.

Furthermore, for the rippled area, by enhancing the contrast of the EDS map (**Fig. R7**), we were able to observe the difference: there is slight Cu enrichment in the rippled area compared to the other three elements, in agreement with the result found in **Fig. R4**. The detailed discussion of the morphological, compositional and electronic properties of the electrode edge and surrounding areas, please refer to our response to comment (4) of Reviewer #1.

Fig. R4. (a) The SEM image of a pristine and a fatigued CIPS capacitor. (b) EDS spectra averaged over different areas marked by the corresponding color boxes in (a). The dash box highlights the stark difference at O K α line. (c) The corresponding oxygen distribution map of (a). (d) EDS line profiles of different elements along the dash line shown in each map. (e) Cu, (f) P, (g) In, and (h) S distribution map of (a).

Fig. R7. (a) AFM topographic image and (b-f) corresponding EDS elemental maps of the rippled area of the fatigued capacitor. (b) O, (c) Cu, (d) In, (e) P, and (f) S map.

To provide complementary information to EDS, we have also conducted depth-profiling XPS to study the compositional variation along the sample depth. As shown in **Fig. R3**, the pristine capacitor possessed a uniform distribution of all four constituent elements for the etching depth from 25 to 200 nm (**Fig. R3a-d**). Besides, almost no oxygen element was detected as expected (**Fig. R3i**). In stark contrast, the chemical composition of the fatigued capacitor varies strongly along the sample depth (**Fig. R3e-h**), which is accompanied by the massive oxygen diffusion into the interior of the crystal (**Fig. R3j**). By integrating the peak areas, we obtained semi-quantitative atomic concentration of each element as a function of the etching depth. Apparently, the elemental distribution of the pristine capacitor is homogeneous from surface to the bulk interior with negligible oxygen amount. The fatigued sample, however, contains substantial oxygen concentrations, which decrease gradually from 40 % at the surface to $\approx 15\%$ at 150 nm depth. Correspondingly, the relative concentrations of In and S are suppressed in the surface layer (depth < 150 nm), but Cu and P exhibit almost constant amount in this region. However, the surface Cu concentration is significantly lower than that of the bulk interior. All the above results support the fact of mass transfer of Cu from the capacitor interior (only the surface layer) towards its edge due to continuous oxidation during electric cycling.

Fig. R3. Depth-resolved XPS study of pristine and fatigued CIPS capacitors. (a-d, i) Depth-profiling of the (a) Cu 2p, (b) In 3d, (c) P 2p, (d) S 2p and (i) O 1s lines of a pristine CIPS capacitor from 25 – 200 nm. (e-h, j) Depth-profiling of the (e) Cu 2p, (f) In 3d, (g) P 2p, (h) S 2p and (j) O 1s lines of a fatigued CIPS capacitor from 25 – 200 nm. Depth profiles of relative atomic concentrations for (k) pristine and (l) fatigued capacitors.

2.The study focuses on the bulk single crystal. What about thin flakes? is there thickness dependence on the fatigue behavior?

Response: We thank the reviewer for bringing out the same interesting question as Reviewer #1. Indeed, we planned to study the thickness dependence on the fatigue behavior. However, there are currently two difficulties that prevent us from measuring few-layer samples. First, the capacitor becomes too leaky for polarization hysteresis loop measurement below 100 nm, and secondly, the area of the exfoliated few-layer flake is usually too small (less than $10 \times 10 \mu\text{m}^2$) for a reliable measurement of the switched polarization. Nevertheless, we still carried out fatigue study on a 172-nm-thick thin flake of CIPS as shown in **Fig. R1**. The fresh capacitor shows well-defined polarization hysteresis loop with a remanent polarization of $\approx 3.5 \mu\text{C cm}^{-2}$. With increasing cycles, the remanent polarization of the thin flake displays similar incremental behavior to that of the bulk sample, until a sudden breakdown close to 10^6 cycles. More efforts are required to investigate the switching and fatigue behavior in the ultrathin limit.

Fig. R1. (a) Optical image of the cross-bar capacitor of a CIPS thin flake. The inset shows the AFM topographic image to determine the flake thickness. (b) Remanent hysteresis loop of the thin flake capacitor by subtracting the non-remnant contribution of the polarization. (c) Polarization hysteresis loops of the thin flake capacitor after cumulative switching cycles. (d) Remanent polarization as a function of the switched cycles.

3. In Fig. 4a, there is an extra shoulder peak by the side of the main peak. What is the origin for this extra peak?

Response: The extra peak originates from the twinning of the single crystal. As the platelike crystal is easy to bend, it is very common to observe multiple twinning peaks during XRD measurements. It should be noted that our previous XRD results were obtained using conventional XRD, which has a large beam size that covers the whole sample volume, including those outside the capacitor. Hence, it may not accurately reflect the structure information of the fatigued area. To gain more insight from the lattice structure of fatigued sample, we have performed Micro X-ray Diffraction (μ XRD). Here, we focused the incident X-ray to $50\ \mu\text{m}$ to detect the structural changes due to fatigue test within an electrode. As shown in **Fig. R6**, the 2D detector allows us to record the conventional 2θ - θ scan and rocking curve along χ direction. In the 2θ - θ pattern, the peak intensities of the fatigued capacitor are reduced

compared to the pristine one, consistent with our previous result. However, this time we didn't observe noticeable peak shift induced by fatiguing. Actually, in our previous XRD results, the shift of (008) peak is tiny (from 56.472° to 56.482°), which translates to a change of only 0.0024 \AA in d spacing of (001) plane. This tiny lattice change is incompatible with the low-polarization to high-polarization phase transition, as proposed previously (Nat. Mater., 19, 43–48, 2020). Therefore, we believe the experimentally observed polarization enhancement is caused by other effect as detailed in the following response. In the χ scan, the pattern of the fatigued sample includes a sharp central peak and a diffused background, which can be attributed to the rippling of the vdW layers. This effect, however, is much weaker within the electrode than in the surroundings, as shown in our previous XRD.

Fig. R6. 2D XRD maps of (a) pristine and (b) fatigued CIPS capacitors around (006) and (008) bragg peaks. (c) 2θ - θ scan of the (006) and (008) peaks of the CIPS capacitors. The inset is the zoomed-in plot of the (008) peak. (d) χ scan of the (008) peak of the CIPS capacitors.

4. The exact PFM measurement conditions for three specimens in Fig. 4 should be provided. In order to directly compare the magnitude of amplitude signals, there should be an identical reference region included during the PFM measurement, since the measuring conditions and cantilever tunings are different. Moreover, in Fig. 4e and f, where amplitude is characterized using PFM, the amplitude contrast (bright/dark) does not necessarily indicate the coexistence

of polarized phases, such as low and high polarization states. Electrostatic forces can significantly influence the PFM results, often causing distinct amplitude contrasts between the upper and lower polarized phases. It is recommended that the author carefully address and eliminate these factors. I recommend the authors follow the recent recipe of "electrostatic blind spot" (<https://arxiv.org/pdf/2112.09665.pdf>). Can the author provide a zoom-in scanning map in the bright and dark amplitude regions?

Response: We totally agree with the reviewer that parasitic effects such as electrostatic force make quantitative PFM very challenging. Special care needs to be taken to minimize these undesirable factors. Following the authors suggestion, we have also tried the so-called "electrostatic blind spot" method to calibrate our PFM system. The result is shown in **Fig. R10** below. Consistent with the literature, we did observe that the measured piezoelectric amplitude is sensitive to the laser spot position. This is possibly due to a change in the inverse optical lever sensitivity (invOLS). Unfortunately, we didn't find a "blind spot" that is insensitive to the applied DC bias (different electrostatic forces). As can be seen in the PFM images, at $V_{dc} = +2/-2$ V, the amplitude signal still decreased to the minimum, accompanied by a 180-degree flip in the phase image, which suggests an overwhelming contribution from the electrostatic force. We speculate that this disagreement with previous report could originate from the much large spot size in our AFM system compared to that in the literature.

To minimize the electrostatic contributions, our protocol in dual AC resonance tracking (DART) PFM measurement is to check the surface potential difference between tip and sample by Kelvin probe force microscopy (KPFM) first, and make sure their potential difference is below 1 V. Stiff probes with large spring constants are widely reported to effectively reduce the electrostatic effect (Sci. Rep. 7, 41657, 2017; Appl. Surf. Sci. 439, 577-582, 2018). Therefore, we carried out quantitative PFM under both resonant-enhanced mode (DART) using a standard probe ($k \approx 2$ N/m) and non-resonant mode using a stiff probe (≈ 40 N/m) for cross-checking. For both methods, the values of invOLS were carefully calibrated using contact force curve beforehand, which translates the AC deflection signal (volt) into the actual displacement (nm) as detected by the probe. In DART method, the measured PFM signals (amplitude, phase, resonant frequency) were fitted using the simple harmonic oscillator (SHO) model, from which the quantitative piezoresponse amplitude can be derived (Nanotechnology 22, 355705, 2011). In non-resonant mode, driving frequency far from the resonance was chosen to prevent signal enhancement (**Fig. R11d, 11e**). The obtained quantitative amplitude images using the two different methods are in good agreement with each other (**Fig. R11a-c**), confirming the reliability of our results.

Fig. R10. (a-h) Dependence of the laser spot position on the piezoelectric response of CIPS sample under different DC bias. Upper panel: the measured amplitude as a function of the DC bias for upward (red) and downward (black) domains. The inset is the corresponding optical image of the laser spot position on the cantilever. Lower panel: PFM amplitude (up) and phase (down) images. The DC bias was continuously changed from -3 to +3 V during the scan as indicated in (a).

Fig. R11. (a) Topographic, (b) quantitative PFM amplitude, (c) PFM phase images of CIPS sample under resonant (upper panel, at 300 kHz) and non-resonant (bottom panel, at 30 kHz) modes. (d) Frequency-dependent piezoresponse amplitude (AC deflection) and phase measured by a stiff probe ($k \approx 40$ N/m). (e) Low-frequency piezoresponse amplitude of a fresh CIPS capacitor as a function of the AC driving voltage. (f) Effective piezoresponse as derived from (e) under different AC voltages.

Based on the method established above, we re-measured the effective piezoresponse of the fresh and fatigued capacitors using quantitative DART-PFM with SHO fitting, as shown in **Fig. R9**. The capacitors were poled into single polarization state first, followed by the exfoliation of the top electrodes to facilitate the PFM imaging. The fresh capacitor exhibits uniform piezoresponse across the scanned area of $30 \times 30 \mu\text{m}^2$, suggesting the high quality of our single crystal sample. In comparison, the fatigued capacitor with enhanced polarization contains regions with suppressed piezoresponse. Besides, scattered small domains with opposite phase signal can be observed. There are two possible explanations for these small domains. It could be the frozen domains, which are not switchable during the pre-poling process. Or, it can be interpreted as the high-polarization phase with positive d_{33} , and consequently opposite phase signal (Nat. Mater. 19, 43–48 2020). With only the PFM results, it is not possible for us to distinguish these two scenarios. However, our updated micro-XRD (**Fig. R6**) results completely ruled out the existence of the high-polarization phase in the fatigued capacitor. Hence, the low-piezoresponse domains are more likely frozen domains or domains with glassy dipoles, which contribute to the dielectric relaxation in the permittivity spectrum.

Majority of the fatigued capacitor shows a piezoresponse comparable to that of the fresh one. By quantitatively measuring the piezoresponse using a stiff probe under non-resonant mode (Fig. R11e, 11f), we obtained the effective d_{33} of the fresh and fatigue sample to be ≈ 20 pm/V, a value much smaller than that reported for CIPS bulk. This is because in current study, the piezoresponse is measured without top electrode. Therefore, the AFM tip only excites a small volume of the sample, which is subjected to strong clamping by the surrounding bulk.

Fig. R9. Optical images of the CIPS single crystal (a) before and (b) after electrode exfoliation. The fresh and fatigued capacitors were poled into single polarization state before the exfoliation. The blue and red boxes indicate the scan areas for fresh and fatigued capacitors, respectively. The white arrows denote markers for positioning. (c-e) Topographic, (f-h) PFM amplitude, and (i-k) PFM phase images of the fresh and fatigued areas as indicated by corresponding color boxes. The PFM was performed in

DART mode using a PtIr-coated probe ($k \approx 2$ N/m). The piezoresponse amplitude was fitted using SHO model to obtain quantitative values.

5. Considering the potential coexistence of low and high polarization phases with different d33 values, it would be more interesting and compelling if the fatigue of d33 and polarization can be investigated.

Response: Referring to our response to the last question, we have excluded the existence of high polarization phase in the fatigued capacitor. The anomalous polarization enhancement is actually caused by the formation of the conductive oxidized layer, which increases the effective electrode area. For the detailed discussion, please refer to our response to comment (4) of Reviewer #1.

6. Compared to previous studies (ACS Applied Materials & Interfaces, 2022, 14(2): 3018-3026), the author should emphasize the importance and significance of their work more prominently.

Response: We need to point out that our work is totally different from the previous study mentioned by the reviewer. The previous study reported the tuning of polarization switching behavior in CIPS capacitor by different DC pulses, while our work focuses on its polarization fatigue and degradation behaviors, which provides important information on the reliability and endurance of device application. The similarity of the two studies is probably the observation of the polarization enhancement. The mentioned study attributed it to the stabilization of high-polarization phase by ionic-migration-induced internal field. However, in our revised manuscript, using multiple microscopy techniques, we clarify that the polarization enhancement in repeatedly cycled capacitor is due to the formation of conductive surface layer that enlarges the total switched area.

7. Some details in the article require careful check. For example, the scale bar length is not indicated in Fig.3. the 100 microns scale bar in Fig. 1 is too small. Furthermore, the experimental methods section lacks sufficient details. It is recommended that the author provide a comprehensive supplementation of these missing details.

Response: We thank the reviewer for pointing out our omission. We have updated the figures and added more information in the experimental methods.

8. The mechanism of induced rippling is illustrated as Vegard effect and ripplocation effect. Can the authors comment on this if the converse flexoelectric effect is also contributing?

Response: We thank the reviewer for the illuminating question. Exactly, due to the inhomogeneous polarization distribution, local strain could be generated by converse flexoelectric effect, which may aggravate the surface rippling of the crystal. We have added the discussion and relevant references in the revised manuscript.

9. In oxide ferroelectrics, one effective way to relieve the polarization fatigue is to use oxide electrodes instead of metal electrodes, so as to suppress the formation of oxygen vacancies. I wonder for CIPS whether we can employ similar strategy. Can the author comment on this issue?

Response: This is a very good suggestion. For CIPS, we expect Cu electrode could partially relieve the problem of Cu loss due to oxidation. However, unlike the robust ferroelectric oxides, sulfides are prone to oxidation and degradation under electrical stressing in air. As a result, surface passivation or encapsulation could be a more effective way to circumvent its fatigue problem.

Summary of main changes (changes in the manuscript are shown in red in the revised version)

1. Added three co-authors: Kazuki Okamoto, Hiroshi Funakubo, Zhongshen Luo, Lixing Kang, who contributed in the new experiments and analyses. Changed Shun Wang to be the co-first author.
2. In the beginning of section 2.1, added some general descriptions of sample information in this work, and included more detailed information in Table S1 in the SI.
3. In the final part of section 2.1, added the fatigue results of CIPS thin flake, and Fig. S5 in the SI.
4. In the first paragraph of section 2.2, revised Figure 2 by adding the line profile analysis of the EDS map and corresponding discussion.
5. In the second paragraph of section 2.2, added new Figure 3 of multiple microscopic imaging results and detailed discussion about the correlation between morphology, composition, structure and property changes in the electrode edge and surrounding rippled area.
6. In the third paragraph of section 2.3, added Fig. S8-S10 in SI and corresponding discussion to clarify the nature of the bubble-like protrusions formed on the electrode.
7. In section 2.3, added new Figure 4 , Fig. S11-S13 in SI, and related discussion of the depth-profiling XPS results, which elucidate the compositional variation across the sample depth.
8. In section 2.4, added new micro-XRD results (Fig. S14 in SI) and PFM results (Fig. S15-S16 in SI).
9. In section 3, rewrote the discussion part on the nature of the bubble-like protrusions and the origin for the polarization enhancement. Added Fig. S18 in SI and the designed experiment to verify the cause for the enhanced polarization.
10. In section 3, added Fig. S19 in SI and discussion to testify the oxidation effect on the fatigue.

REVIEWERS' COMMENTS

Reviewer #1 (Remarks to the Author):

In the revision, the authors have added sufficient control experiments and discussion to support their findings. The careful revisions are appreciated. Polarization fatigue has been attributed to ion migration and surface oxidation. In the revised version, the authors made position-dependent XPS measurements in the z direction, which provided some direct information about surface oxidation. Unfortunately, for ion migration, no direct evidence of Cu⁺ ions entering the vdW gap has been directly observed. Direct observation of Cu⁺ migration or assistance with some first-principles calculations may help provide a clearer understanding of ion migration, but this will not prevent the publication of this manuscript. One last thing is that we can find that the polarization coercive field in ~200 nm flakes is an order of magnitude larger than in bulk samples, and the authors may need to discuss this.

Reviewer #2 (Remarks to the Author):

The authors addressed well the issues I concerned and I recommend it to be published on Nature Communications.

Response to Reviewer #1

REVIEWER COMMENTS

Reviewer #1 (Remarks to the Author):

In the revision, the authors have added sufficient control experiments and discussion to support their findings. The careful revisions are appreciated. Polarization fatigue has been attributed to ion migration and surface oxidation. In the revised version, the authors made position-dependent XPS measurements in the z direction, which provided some direct information about surface oxidation. Unfortunately, for ion migration, no direct evidence of Cu⁺ ions entering the vdW gap has been directly observed. Direct observation of Cu⁺ migration or assistance with some first-principles calculations may help provide a clearer understanding of ion migration, but this will not prevent the publication of this manuscript. One last thing is that we can find that the polarization coercive field in ~200 nm flakes is an order of magnitude larger than in bulk samples, and the authors may need to discuss this.

Response: We thank the reviewer for his valuable suggestions for the manuscript and positive assessment of our revision. We agree that further theoretical calculation may provide more insight into a better understanding of the ion migration process during the fatigue process.

Regarding to the last question on the coercive field, coercive voltage of ~200 nm flake is about 2 V, which translates into a coercive field of ~ 100 kV/cm, about 4 times as that of the bulk crystal (~ 15 μm). The increase of coercive field (E_c) with reducing thickness (d) is a common feature of ferroelectric materials, which can be empirically described by Kay-Dunn law, $E_c \propto d^{-2/3}$ (H. Kay, J. Dunn, *Philos. Mag.* 1962, **7**, 2027). We have added the discussion in the Supplementary information.

Response to Reviewer #2

Reviewer #2 (Remarks to the Author):

The authors addressed well the issues I concerned and I recommend it to be published on Nature Communications.

Response: We thank the reviewer for his/her valuable comments and suggestions in improving the manuscript.